

# Innovative road distress detection (IR-DD): an efficient and scalable deep learning approach

Ahsan Zaman Awan[1], Jiancheng (Charles) Ji[2], Muhammad Uzair[1], Irshad Ullah[1], Waqar Riaz[2,3] and Tao Gong[2]

[1] School of Computer Science and Engineering, Central South University, Changsha, Hunan, China
[2] Institute of Intelligent Manufacturing, Shenzhen Polytechnic University, Shenzhen, Guangdong, China
[3] Shenzhen Institutes of Advanced Technology, Chinese Academy of Sciences, Shenzhen, Guangdong, China

Corresponding author
Jiancheng (Charles) Ji,
jcji20@szpu.edu.cn

## ABSTRACT

In the rapidly evolving landscape of transportation infrastructure, the quality and condition of road networks play a pivotal role in societal progress and economic growth. In the realm of road distress detection, traditional methods have long grappled with manual intervention and high costs, requiring trained observers for time-consuming and expensive data collection processes. The limitations of these approaches are compounded by challenges in adapting to diverse road surfaces and handling low-resolution data, particularly in early automated distress survey technologies. This article addresses the critical need for efficient road distress detection, a key component of ensuring safe and reliable transportation systems. Effectively addressing these challenges is crucial for enhancing the efficiency, accuracy, and safety of road distress detection systems. Leveraging advancements in object detection, we introduce the Innovative Road Distress Detection (IR-DD), a novel framework that integrates the YOLOv8 algorithm to enhance the accuracy and real-time capabilities of road distress detection, catering to applications such as smart cities and autonomous vehicles. Our approach incorporates bidirectional feature pyramid network (BiFPN) recursive feature fusion and bidirectional connections to optimize the utilization of multi-scale features, addressing challenges related to information loss and gradients encountered in traditional methods. Comprehensive experimental analysis demonstrates the superior performance, efficiency, and robustness of our integrated approach, positioning it as a cost-effective and compelling alternative to conventional road distress detection methods. Our findings demonstrate the superior performance of our approach compared to other state-of-the-art methods across various evaluation metrics, including precision, recall, F1 score, and mean average precision (mAP) at different intersection over union (IoU) thresholds. Specifically, our method achieves notable results with a precision of 0.666, F1 score of 0.630, mAP@0.5 of 0.650, all while operating at a speed of 86 frames per second (FPS). These outcomes underscore the effectiveness of our approach in real-time road distress detection. This article contributes to the ongoing innovation in object detection techniques, emphasizing the practicality and effectiveness of our proposed solution in advancing the field of road distress detection.

# INTRODUCTION

In today's rapidly evolving world, efficient and safe transportation infrastructure plays a pivotal role in ensuring societal progress and economic growth. The quality and condition of road networks are not only essential for the seamless movement of people and goods but also significantly impact road safety, vehicle maintenance costs, and overall economic vitality. However, the long-term sustainability and viability of transportation infrastructure hinge on the proactive management, preservation, and rehabilitation of road pavements. Advanced pavement distress detection methods are essential for timely identification and resolution of issues like cracking and surface defects. Leveraging cutting-edge technologies such as 2D high-resolution cameras, convolutional neural networks (CNN), and automated imaging systems, these methods accurately assess pavement conditions, contributing to informed decision-making in infrastructure maintenance (*Nguyen et al., 2023*). By overcoming the limitations of manual inspection, they offer efficient, automated solutions, enhancing the resilience and longevity of transportation networks.

Object detection, a vital aspect of computer vision, entails identifying and pinpointing objects in images or videos. The primary goal is to precisely locate and classify objects, assigning them the appropriate labels. This versatile technique finds applications in various domains such as predicting stock values (*Rather, Agarwal & Sastry, 2015*) intrusion detection (*Kim et al., 2016*), landslide detection (*Mezaal et al., 2017*), gene expression (*Quang & Xie, 2016*), data handling (*Liao et al., 2017*), aspect-based sentiment analysis (*Sirisha & Bolem, 2022*), and generating captions from videos (*Xu et al., 2018*). Specifically, in the context of road distress detection, object detection models leverage deep learning architectures and algorithms to identify and categorize road-related anomalies and issues in real-world scenarios.

In recent years, significant progress has been made in object detection, particularly through the evolution of deep learning techniques, which can be broadly categorized into two stages. Firstly, the two stage object detection involves an initial object region proposal phase, followed by object classification and bounding box regression. Notably, detectors following this approach, including R-CNN (*Girshick et al., 2014*), Faster-RCNN (*Ren et al., 2015*), and Mask-RCNN (*Zhang, Chang & Bian, 2020*) algorithms, exhibit higher accuracy despite being relatively slower. On the other hand, one stage object detection skips the object region proposal step and directly predicts bounding boxes from images, resulting in faster processing. However, these detectors may struggle with detecting smaller objects. Noteworthy examples of single stage detectors, such as Single Shot Multibox Detector (SSD) (*Liu et al., 2016*), You Only Look Once (YOLO) (*Liu et al., 2018a*), EfficientDet (*Tan, Pang & Le, 2020*), are known for their rapid inference speed, making them practical for various applications. In recent research endeavors, the need for automated detection of road pavement damages has been addressed, employing a tire noise propagation-based

multi-modal approach (*Li et al., 2024*). The proposed method introduces innovative features, demonstrating superior performance in robustness and generalizability. The proposed scheme (*Wen et al., 2024*), including PCD software for real-time crack detection, aligns with the overarching goal of enhancing road infrastructure management for safety, efficiency, and environmental responsibility.

Effectively representing and processing multi-scale features poses a significant challenge in object detection. Early detectors often relied on making predictions directly from the pyramidal feature hierarchy extracted from backbone networks (*Liu et al., 2016*; *Cai et al., 2016*; *Sermanet et al., 2013*). The feature pyramid network (FPN) (*Lin et al., 2017*) introduced a top-down pathway to integrate multi-scale features, pioneering this approach. Subsequent advancements include PANet (*Liu et al., 2018b*), which added a bottom-up path aggregation network to FPN, STDL (*Zhou et al., 2018*), introducing a scale-transfer module for cross-scale feature utilization, M2det (*Zhao et al., 2019*), proposing a U-shape module for multi-scale feature fusion, and G-FRNet (*Amirul Islam et al., 2017*), which incorporated gate units to control information flow across features. More recently, NAS-FPN (*Ghiasi, Lin & Le, 2019*) employed neural architecture search to automatically design the topology of the feature network. While achieving superior performance, NAS-FPN demands thousands of GPU hours during the search process, and the resulting feature network can be irregular and challenging to interpret.

In this article, we introduce Innovative Road Distress Detection (IR-DD), a novel framework that utilizes the You Only Look Once version 8 (YOLOv8) model, enhancing accuracy while enabling real time detection—crucial for applications such as smart cities and autonomous vehicles. YOLOv8, the latest addition to the YOLO series released on January 10th, 2023, demonstrates improved speed and accuracy compared to its predecessor. It employs an anchor-free architecture, simplifying training across various datasets without relying on anchor boxes (*Jocher, Chaurasia & Qiu, 2023*). This progression underscores the continual refinement and innovation within the YOLO series, making notable contributions to real-time object detection in diverse applications. We also incorporate Bidirectional Feature Pyramid Network (BiFPN) for recursive feature fusion and bidirectional connections, optimizing the utilization of multi-scale features. BiFPN strives to enhance the fusion of multi-scale features through a more intuitive and principled optimization approach. Our road distress detection model's peak performance is evident through thorough experimental analysis. The integration of YOLOv8 and BiFPN in our object detection framework signifies a substantial advancement in road distress scenarios. By leveraging the strengths of BiFPN and YOLOv8 model, our approach achieves superior performance, enhancing accuracy, real-time capabilities, and overall efficiency in detecting anomalies with precision and speed. The seamless fusion of YOLOv8's object detection capabilities with BiFPN's feature fusion optimizes multi-scale feature capture, addressing the complexities of road distress detection tasks effectively. Our comprehensive evaluation demonstrates the effectiveness of this integrated approach in minimizing information loss, preserving critical details, and setting a new standard for intelligent transportation systems focused on advanced object detection in road distress scenarios.

The following are the key contributions of this article:

- In this article introduces IR-DD, an innovative framework for road distress detection, utilizing the YOLOv8 algorithm. This novel object detection approach enhances accuracy while enabling real time detection, crucial for applications like smart cities and autonomous vehicles. The cost-effective nature of our proposed method positions it as a compelling alternative to traditional road distress detection methods.

- This novel approach, incorporating BiFPN recursive feature fusion and bidirectional connections, optimizes the utilization of multi-scale features. By addressing information loss and gradients, traditionally encountered in FPN, our method enhances the robustness of feature representations, resulting in improved efficiency and effectiveness for object detection models.

- Our road distress detection system's optimal performance is rigorously demonstrated through comprehensive experimental analysis. The integration of YOLOv8 and BiFPN not only yields superior accuracy and real-time capabilities, as evidenced by our results but also underscores the efficiency and robustness of our approach. The comprehensive analysis further emphasizes its efficacy in mitigating information loss. This empirical evidence solidly supports the assertion that our integrated approach stands as an optimal and highly effective solution for advanced object detection in road distress scenarios.

The remainder of this article is organized as follows. "Related Work" undertakes a comprehensive review of pertinent literature, offering insights into prior work in the field. Moving forward, "Proposed Method" elucidates the proposed approach, providing a detailed explanation of the methodology. "Implementation and Evaluation" is dedicated to presenting and analyzing the experimental results. In "Conclusion and Future Works", the article concludes by summarizing key findings and delineates potential directions for future research.

## RELATED WORK

The You Only Look Once (YOLO) series of object detection algorithms has played a crucial role in advancing real-time detection capabilities. These algorithms have been transformative by introducing a pioneering single-shot detection approach. This approach involves processing an entire image in a single pass through a convolutional neural network (CNN). Notably, it eliminates the need for separate stages of region proposal generation and object classification, significantly enhancing inference times. The success of YOLO algorithms extends across various applications, particularly in the field of self-driving cars. The inaugural YOLO algorithm, YOLOv1, was introduced in 2015 (*Redmon et al., 2016*). Comprising 24 convolutional layers and two fully connected layers, YOLOv1 demonstrated unprecedented speed, outperforming contemporaneous object detectors like Fast R-CNN. Despite its real-time capabilities, YOLOv1 faced challenges in accurately detecting and localizing small objects.

In 2016, YOLOv2 addressed these limitations, offering enhanced accuracy while maintaining rapid inference speeds (*Redmon & Farhadi, 2017*). Subsequently, YOLOv3

aimed for heightened accuracy through the integration of larger CNNs, employing a
Darknet-53 backbone with 53 convolutional layers (*Redmon & Farhadi, 2018*). YOLOv4,
proposed by *Bochkovskiy, Wang & Liao (2020)*, further advanced the algorithm's structure
and optimization methods, incorporating the CSPDarknet-53 backbone for improved
efficiency and accuracy (*Bochkovskiy, Wang & Liao, 2020*). YOLOv5, introduced in the
same year, prioritized speed without compromising accuracy. While similar to YOLOv4,
YOLOv5 adopted the PyTorch framework instead of DarkNet (*Jocher et al., 2020*).
YOLOv6 (2022) introduced advancements to the network structure by incorporating
RepVGGEfficientRep and Rep-PAN as the foundation. Utilizing an Efficient Decoupled
Head, YOLOv6 optimized object detection efficiency without compromising accuracy (*Li
et al., 2022*). Released in July 2022, YOLOv7, built upon the YOLOv4 architecture,
introduced enhancements through the Extended Efficient Layer Aggregation Network (E-
ELAN), surpassing its predecessors in both speed and accuracy (*Wang, Bochkovskiy &
Liao, 2023*).

Two-stage algorithms follow a process wherein they initially generate a set of candidate
bounding boxes as samples and subsequently classify these samples using a convolutional
neural network (CNN). Notable examples of such algorithms encompass Fast R-CNN
(*Girshick, 2015*), Faster R-CNN (*Ren et al., 2015*), Cascade RCNN (*Cai & Vasconcelos,
2018*), and others. *Nie & Wang (2018)* introduced a crack detection model based on Faster
R-CNN, leveraging transfer learning through parameter fine-tuning to identify various
pavement issues like cracks, looseness, and deformation. *Hascoet et al. (2020)* employed
Faster-RCNN for crack detection, improving detection performance through techniques
like label smoothing, and shared insights into deploying their model on local road
networks. In contrast, one-stage algorithms approach object detection as a regression task,
directly predicting bounding boxes and categories for multiple locations across the entire
image. Examples of one-stage algorithms include SSD (*Redmon & Farhadi, 2018*), the
YOLO series (*Bochkovskiy, Wang & Liao, 2020*; *Wang, Bochkovskiy & Liao, 2023*),
Centernet (*Duan et al., 2019*), and EfficientDet (*Tan, Pang & Le, 2020*).

In this article (*Wang et al., 2024*), the introduction of SwinCrack, a groundbreaking
pavement crack detection model built on the Swin-Transformer architecture, is presented.
SwinCrack effectively overcomes the drawbacks of CNN-based networks, achieving
enhanced accuracy and efficiency in crack detection. Leveraging modules such as CPEL,
CSTB, DFN, and CAGSC, SwinCrack excels in capturing spatial context and fine-tuning
crack boundaries. This work signifies a notable advancement in automated pavement crack
detection methodologies. Furthermore, recent research has presented a groundbreaking
approach to detecting renal cell hydronephrosis using a novel deep convolutional neural
network (DCNN) architecture (*Islam et al., 2024*). The study evaluates the DCNN against
established models (VGG16, ResNet50, InceptionV3), demonstrating its superior
performance. Key contributions include the innovative DCNN design, comprehensive
architecture comparisons, and detailed reproducibility information. This research not only
advances hydronephrosis detection but also underscores the broader potential of deep
learning in medical diagnostics. In this article (*Xu et al., 2024*) the authors addresses
challenges in tunnel crack width identification, focusing on a large subway tunnel and

utilizing a tunnel rail inspection car equipped with industrial cameras. The proposed method integrates laser rangefinders for precise measurements, corrects three-dimensional cracks, and employs the YOLOv8 algorithm for intelligent extraction of crack morphology. Results show close alignment between YOLOv8-based crack detection and manual methods. The approach, using a tunnel inspection vehicle and YOLOv8, proves feasible for accurate crack recognition on the tunnel tube sheet, offering potential applications and serving as a valuable reference for crack assessments.

In the realm of road distress detection, previous research has laid the groundwork for innovative approaches. Notably, advancements in object detection have been witnessed with the integration of sophisticated models such as YOLOv8 (You Only Look Once version 8). YOLOv8 has proven instrumental in achieving superior accuracy and real-time capabilities, making it a prominent choice in the field. The significance of leveraging state-of-the-art technologies, such as YOLOv8, is evident in enhancing the accuracy and efficiency of road distress detection systems.

## PROPOSED METHOD

In this article, we present our IR-DD deep learning approach, designed to address the challenges of accuracy, real-time performance, and scalability in road distress detection. To achieve this balance, we leverage YOLOv8, a cutting-edge one-stage and one-scale object detection model (*Talaat & ZainEldin, 2023*) renowned for its accuracy and efficiency. YOLOv8 signifies a noteworthy advancement in the realms of object detection, image categorization, and instance segmentation. Developed by Ultralytics, it streamlines the developer experience and introduces a key change by directly predicting object centers instead of anchor-box offsets. This innovation accelerates Non-Maximum Suppression, simplifies box configuration forecasting, and brings changes to the convolutional structure, including modifications to building blocks and kernel sizes. YOLOv8 also reduces tensor size and parameters through direct neck feature concatenation, improving accuracy. Furthermore, it incorporates mosaic augmentation, enhancing the model's ability to detect objects in different environments and handle partial occlusions. To optimize feature extraction, we leverage the EfficientNet backbone (*Tan & Le, 2019*). Our approach commences with feature extraction from input video/images, focusing on critical elements like crack orientation, background context, brightness variations, and distress area extent. Figure 1 illustrate the YOLOv8 general architecture (*Solawetz & Francesco, 2023*). The integration of BiFPN enhances multi-scale feature fusion, particularly beneficial for accurately detecting small defects in complex visual backgrounds. These advantages collectively position YOLOv8 and BiFPN as a powerful and balanced combination, addressing the unique demands of speed, accuracy, and flexibility in road distress detection.

In this comprehensive exploration, our article unfolds the intricacies of our proposed model across various pivotal phases. "The IR-DD Model System" meticulously details the IR-DD deep learning approach, focusing on its core structure and innovative features designed to address challenges in accuracy, real-time performance, and scalability in road distress detection. Transitioning to "The IR-DD Model Algorithm", we delve into the

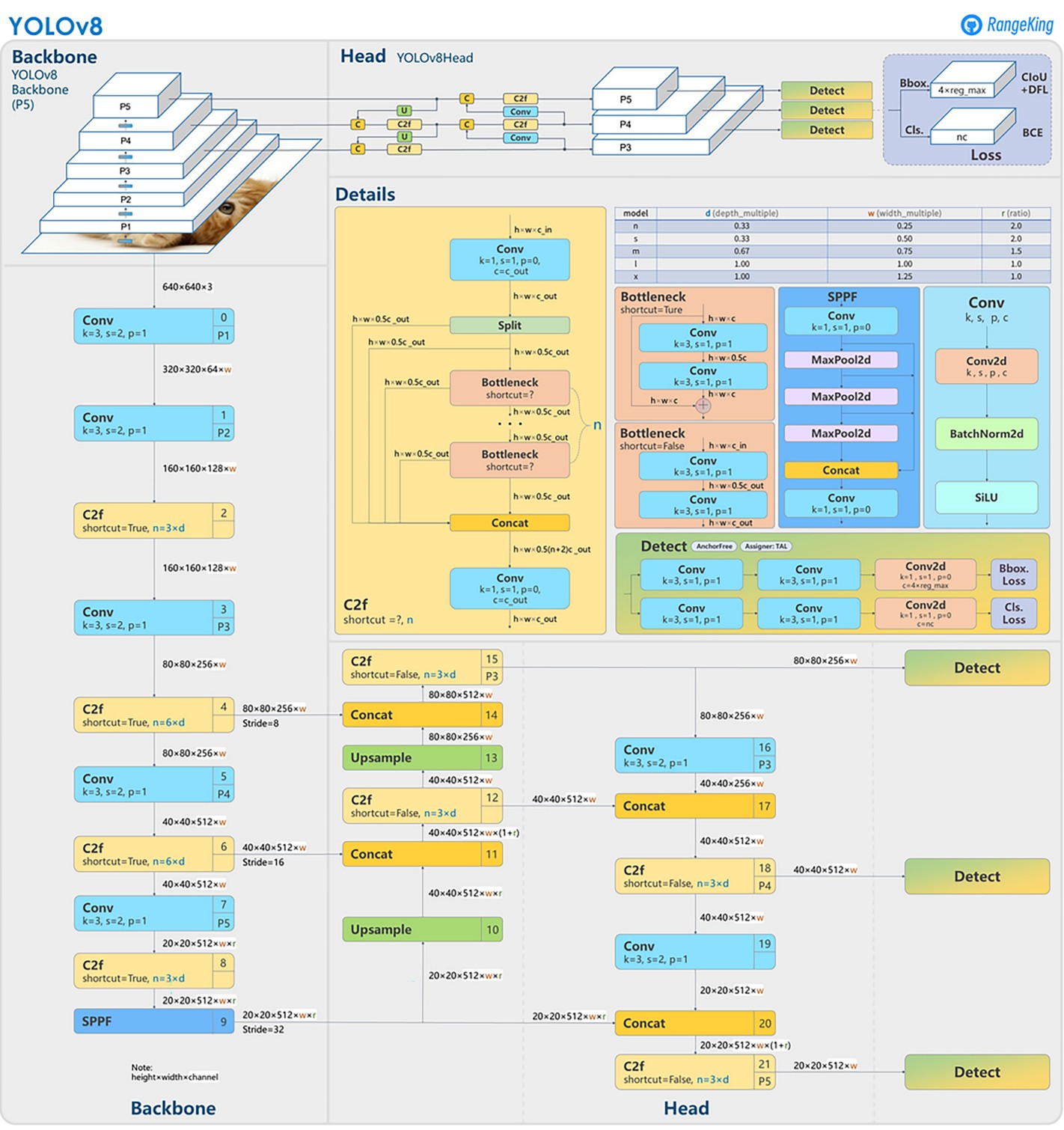

**Figure 1  YOLOv8 architecture.**

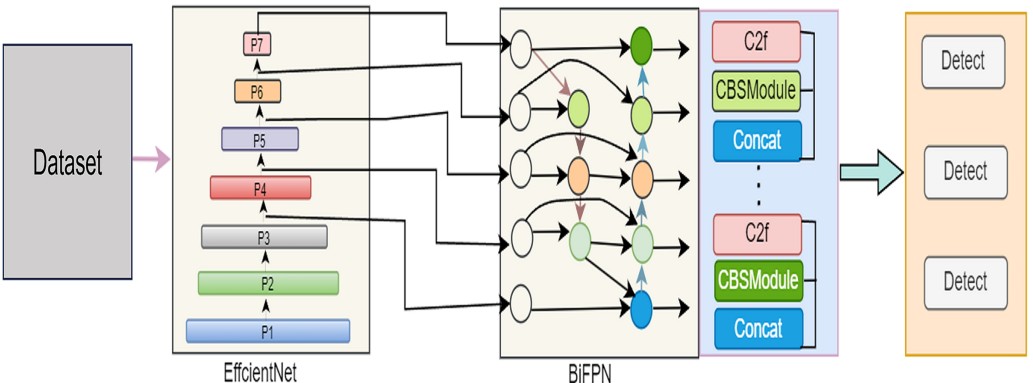

**Figure 2 Architecture of the proposed IR-DD model.**

IR-DD algorithm, outlining its algorithmic underpinnings, efficiency, and precision in object detection and image categorization. "Bidirectional Feature Pyramid Network" sheds light on the BiFPN, emphasizing its role in optimizing feature fusion and enhancing multi-scale features for accurate road distress detection. Finally, "Anchor-Free Detection" elucidates the intricacies of anchor-free detection, highlighting its innovative aspects in achieving accurate and efficient distress detection in diverse environments. Figure 2 illustrate the proposed model architecture.

## The IR-DD model system

The IR-DD deep learning approach leverages the YOLOv8 detection model, renowned for its swift and accurate object detection capabilities, all achieved without the necessity of a regional proposal network. This system undergoes optimization to streamline the parameter count required for detection, enhancing its overall efficiency. Employing computer vision techniques, the IR-DD deep learning approach autonomously identifies road distress anomalies within images and video streams. Data preprocessing and object detection are the essential phases of the IR-DD model.

### Data preprocessing

In data preprocessing phase, data collection and preparation are integral phases. Data collection involves acquiring a diverse dataset containing images and videos showcasing road distress anomalies, along with images lacking such anomalies. Stringent curation procedures, including duplicate removal and precise labeling, are necessary. Labeling can be performed manually or aided by automated tools, with a focus on categorizing images as containing or lacking road distress anomalies. Maintaining a balanced dataset, with an equitable distribution of both types of images, is crucial to prevent model bias.

Data preparation is equally vital in preparing the dataset for training and testing the road distress detection system. The process commences with the meticulous labeling of images and videos, where bounding boxes are delineated around road distress anomalies. This can be accomplished through manual annotation or labeling tools such as LabelImg. The annotated data is subsequently partitioned into distinct training and testing subsets, ensuring they faithfully represent the broader dataset. Additional preprocessing steps, such

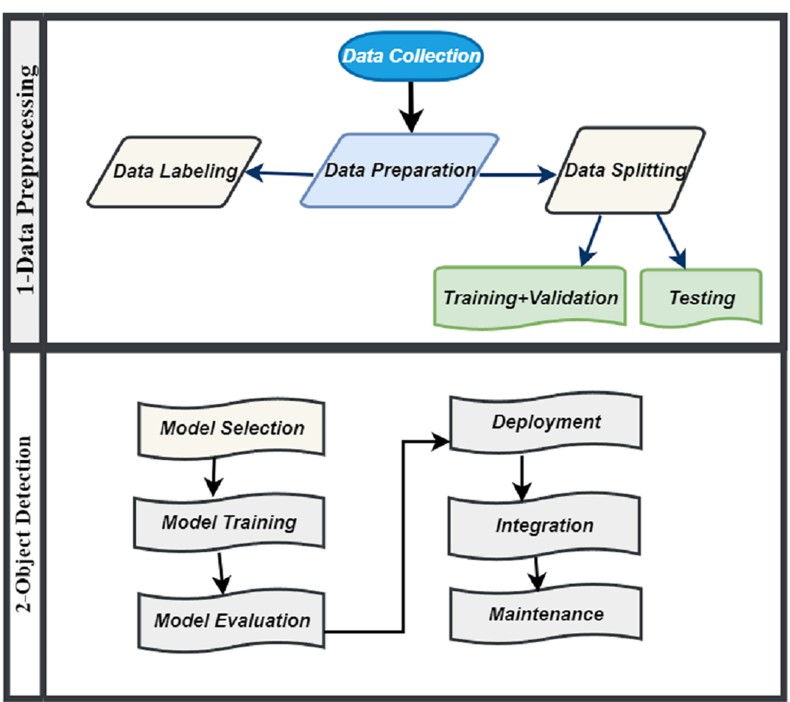

**Figure 3** **Methodology of IR-DD model system.**

as resizing or normalization, may be applied to enhance data consistency. The ultimate goal is to cultivate a substantial and balanced dataset capable of robust generalization to novel data instances.

### Object detection

Object detection involved several stages like model selection, training, evaluation, deployment, integration and maintenance which is illustrated in Fig. 3. In object detection phase, we initiate the model development process by thoughtfully selecting an object detection algorithm for training. We evaluate various options, including Faster R-CNN, SSD and YOLOv8, each having its unique set of advantages and limitations. However, we have selected YOLOv8 as the algorithm for our model. YOLOv8 is highly esteemed for its capability to achieve a harmonious balance between speed and accuracy, aligning seamlessly with the particular requirements of our system. Following the training phase, we evaluate the model's performance using crucial metrics such as accuracy, precision, recall, and F1 score. In the event that the model does not meet expectations, we implement fine-tuning by adjusting hyperparameters or enhancing the dataset. This iterative process aims to strike a balance between overfitting and underfitting, ultimately refining the model's accuracy and reliability.

Deployment is a pivotal phase that entails integrating the trained model into a real-time system for processing live video streams from road cameras. This process demands significant computing power and GPU support to facilitate real-time analysis. The system identifies road distress anomalies and utilizes a confidence threshold to manage false positives, ensuring precise and reliable detection. Integration is another pivotal stage where

| Algorithm 1  IR-DD algorithm. |
|---|
| Input: Dataset |
| Output: Detected Objects |
| Steps: |
| 1: Initiate the input source for video/image (pre-recorded video/image file or live camera). |
| 2: Start the video/image capture process. |
| 3: **while** there are frames in the video/image **do** |
| 4:     Implement image pre-processing techniques on the current frame. |
| 5:     Send the pre-processed frame to the YOLOv8 model for object detection |
| 6:     Examine the identified objects for classes related to road distress, such as cracks, potholes, and surface damage. |
| 7:     **if** A class associated with road distress is identified **then** |
| 8:         Trigger an alarm. |
| 9:         Notify the relevant authorities. |
| 10:    **end if** |
| 11:    Save the resulting video/image, emphasizing the identified road distress anomalies. |
| 12: **end while** |
| 13: Halt the process of capturing video or image. |

we seamlessly merge the road distress detection system with complementary systems like emergency response, traffic management, and maintenance alerts. This integration enables rapid responses, including road closures and maintenance alerts, ultimately optimizing road safety and response efficiency. Maintenance plays an ongoing role in upholding the effectiveness of the deployed system. It involves continuous model enhancement, routine testing, and meticulous upkeep of hardware and software components. These efforts ensure the system's ongoing reliability and adaptability to changing conditions, all contributing to the promotion of road safety.

In the IR-DD model, our approach utilizes computer vision to swiftly identify road distress anomalies in real-time, whether from live camera feeds or pre-recorded video and image files. As depicted in Algorithm 1, it leverages a pre-trained YOLOv8 object detection model, trained on a diverse dataset that includes both road distress and non-distress images.The model functions by taking a dataset of video/image frames as input and generating outputs that identify anomalies, covering diverse road distress classes such as cracks, potholes, or surface damage. This model systematically processes each frame within the video stream. Before forwarding frames to the YOLOv8 model for object detection, the system initiates image pre-processing techniques customized for the current frame. If the model identifies any road distress anomaly, it promptly triggers an alarm and notifies the relevant authorities without delay. Furthermore, it archives the output video/image, duly emphasizing the detected road distress anomalies. The IRDD approach stands as a robust solution for real-time road distress detection, fostering swift and effective responses to potential road hazards.

## The IR-DD model algorithm

The IR-DD algorithm, driven by YOLOv8, integrates cutting-edge technology with systematic workflow to enhance road safety by rapidly identifying and reporting road distress anomalies. Its systematic approach ensures that distress situations are identified, addressed, and recorded efficiently, making it a valuable tool in the maintenance and management of road infrastructure.

The IR-DD algorithm, leveraging the high-performance YOLOv8 object detection model, unfolds through a meticulously orchestrated sequence of stages, each contributing to the overall efficacy of the system.

- **Defining the video/image source:** The initial stage involves the specification of the video/image source, whether it be a live camera feed capturing real-time data or a pre-recorded video/image file. This crucial step ensures that the system is appropriately aligned with the intended application scenario.
- **Video/image frame capture:** With the video/image source established, the system commences the video/image capture process. Frame by frame, it systematically processes each segment of the video/image, ensuring that no pertinent information is overlooked.
- **Image pre-processing:** Prior to subjecting each video/image frame to object detection, the system employs a suite of image pre-processing techniques. These techniques aim to enhance the quality and utility of the input data, which is essential for accurate and robust detection. The pre-processed frame is then forwarded to the YOLOv8 model for comprehensive object detection.
- **Object classification:** The YOLOv8 model plays a pivotal role in this stage. It rigorously examines each frame, categorizing detected objects into various classes. In the context of road distress detection, these classes typically include cracks, potholes, and surface damage. This classification process is a critical component of the system, as it enables the precise identification of distress-related anomalies.
- **Alert triggering:** The system is designed to respond promptly and decisively. In the event of detecting an object belonging to a road distress-related class, such as a crack or pothole, the system triggers an alarm. This alarm serves as a real-time notification mechanism, promptly relaying the information to the relevant authorities. This immediate response ensures swift and targeted action to address road anomalies and enhance safety.
- **Video/image capture conclusion:** As the video processing nears its conclusion, the system meticulously conserves the output video/image. In this recorded video/image, the detected road distress anomalies are distinctly highlighted. This visual representation is invaluable for post-event analysis, road maintenance planning, and record-keeping, offering a comprehensive record of the distress occurrences.

## Bidirectional feature pyramid network

The incorporation of Bidirectional Feature Pyramid Network (BiFPN) into the YOLOv8 model for road distress detection significantly elevates the model's capacity to capture

multi-scale features, thereby enhancing its efficacy in detecting road anomalies. In the realm of road maintenance and safety, the accurate identification of distress anomalies such as cracks and potholes is pivotal for ensuring road quality and driver safety. The integration of BiFPN into the YOLOv8 framework equips the model with the ability to efficiently analyze features at different scales, enabling precise detection of distress anomalies across various road conditions and sizes. BiFPN's bidirectional connections within the feature pyramid network facilitate seamless information flow both upward and downward through the network layers. This bidirectional flow enhances the model's contextual understanding of road scenes, allowing it to discern subtle distress patterns and variations in road surfaces. The synergistic integration of YOLOv8 and BiFPN empowers the model to dynamically adapt to diverse road environments, ensuring robust performance in detecting anomalies irrespective of scale or complexity.

Furthermore, the integration of BiFPN not only enhances the accuracy of distress detection but also improves the efficiency of real-time anomaly identification. This advancement in intelligent transportation systems holds great promise for enhancing road maintenance practices, optimizing safety measures, and ultimately contributing to smoother and safer driving experiences for all road users. Algorithm 2 illustrates the YOLOv8 model enabled with BiFPN.

Mathematically, the BiFPN process, when integrated into YOLOv8 for road distress detection, can be represented as follows:

$$P_i^{\text{up}} = P_{i-1} + \text{UpSample}(P_i^{\text{down}}) \tag{1}$$

$$P_i^{\text{out}} = \text{ReLU}(\text{Conv}(P_i^{\text{up}})). \tag{2}$$

Here, $P_i^{\text{up}}$ represents the upsampled feature map from the lower-level pyramid, $P_i^{\text{down}}$ is the feature map from the higher-level pyramid, and $P_i^{\text{out}}$ denotes the fused feature map. This fusion process, when applied within the YOLOv8 model, ensures that distress anomalies on the road are detected accurately and reliably, contributing to improved road safety and maintenance efforts. The YOLOv8-BiFPN model, optimized for Road Distress Detection, can efficiently identify and localize road anomalies, providing an invaluable tool for infrastructure management and maintenance.

Following steps are follow for integrating BiFPN into YOLOv8 model for road distress detection:

- **Data preparation:** Prepare a dataset of road distress images with annotations (*e.g.*, bounding boxes for distress anomalies).
- **Backbone feature extraction:** Utilize a pre-trained or custom backbone (EffcientNet) to extract features from the input images.
- **BiFPN integration:** Incorporate the BiFPN module into the YOLOv8 architecture after the backbone network. Configure the BiFPN to fuse features from different pyramid levels bidirectionally.
- **Feature fusion:** Implement feature fusion within the BiFPN to capture multi-scale features efficiently.

| **Algorithm 2** | **YOLOv8 enabled bidirectional feature pyramid network (BiFPN) algorithm.** |
|---|---|

Steps:

1: Initialization of the YOLOv8 model: Model = YOLOv8()

2: Loading the backbone:

    Backbone = load_retrained_backbone()

3: Adding the BiFPN module:

    BiFPN = BidirectionalFPN()

4: Initializing the object detection head:

    Head = ObjectDetectionHead()

5: Connecting layers:

    Model = connect_layers(Backbone, BiFPN, Head)

6: Defining the loss function:

    loss = calculate_loss(predictions, targets)

7: Preparing the dataset: Data preparation and loading into data loaders.

8: Training loop: Iterating over the dataset and optimizing the model parameters using backpropagation.

9: Inference: Detections = Model.detect(image)

10: Evaluation:

    Evaluation_results = Evaluate(Model, test_dataset)

11: Fine-tuning: Optionally fine-tuning and optimizing hyperparameters.

12: Saving the model:

    save_model(Model,$'$ road_distress_detection_model.pth$'$)

- **Object detection head:** Follow the BiFPN with an object detection head. Design the head to predict bounding boxes, object confidence scores, and class probabilities.
- **Training:** Train the YOLOv8-BiFPN model on the road distress dataset, using ground truth annotations. Use loss functions to optimize the model.
- **Inference:** Apply the trained YOLOv8-BiFPN model for road distress detection on new images.
- **Post-processing:** Implement post-processing steps like non-maximum suppression to filter and refine the detection results.
- **Evaluation:** Assess the model's performance by utilizing metrics such as precision, recall, F1-score, and mean average precision (mAP).
- **Fine-tuning and optimization:** Optionally, fine-tune hyperparameters and model architecture to achieve optimal performance.

## Anchor-free detection

This section provides a comprehensive exploration of anchor-free model architecture, elucidating its significance in the realm of object detection tasks. Understanding how the anchor-free model is structured and carefully adjusting our model's settings is a key

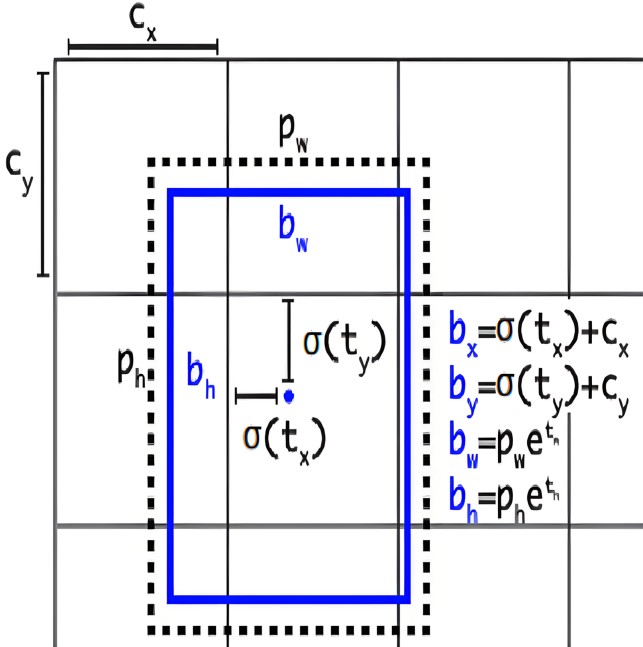

**Figure 4  Graphical representation of anchor boxes in object detection.**

strategy to improve the performance of our object detection model, especially when dealing with objects that have irregular shapes. YOLOv8 adopts an anchor-free model architecture, deviating from the conventional approach of predicting offsets from predefined anchor boxes. Instead, it directly predicts the center of an object, eliminating the reliance on anchor boxes for bounding box predictions. To visually represent the concept of anchor boxes in object detection, Fig. 4 illustrates the graphical depiction of anchor boxes (*Solawetz & Francesco, 2023*). These boxes are pre-defined bounding boxes with varying scales and aspect ratios, serving as crucial elements in object detection algorithms for anchoring and predicting objects within an image. The network generates predictions for five bounding boxes at each cell of the output feature map. For each bounding box, the network predicts five coordinates: $t_x$, $t_y$, $t_w$, $t_h$, and $t_o$. If the cell is displaced from the top-left corner of the image by the offset ($c_x$, $c_y$), and the bounding box prior has width ($p_w$) and height ($p_h$), then the predictions are as follows:

$$b_x = \sigma(t_x) + c_x$$
$$b_y = \sigma(t_y) + c_y$$
$$b_w = p_w \cdot e^{t_w}$$
$$b_h = p_h \cdot e^{t_h}$$
$$\text{Pr(object)} \cdot \text{IOU}(b, \text{object}) = \sigma(t_o)$$

Object detection, distinct from image classification, grapples with the challenge of identifying and precisely localizing multiple objects, potentially belonging to various classes, within a single image. The primary objective is to accurately predict the presence

and location of all these objects. The subsequent steps outline the process involved in anchor-free detection.

- **Center points prediction:** In anchor-free methods, the network predicts the center points of potential objects. This is often represented as a heat map where each point in the heat map corresponds to a potential object center.
  Mathematically:
  Let us denote the heat map for object center points as $C$, and $C(x, y)$ represents the confidence score of an object center at location $(x, y)$.

- **Bounding box regression:** Once the center points are predicted, anchor-free methods perform bounding box regression relative to these centers. They predict the offset $(\Delta x, \Delta y)$ for each object center, which determines the position of the bounding box around that center.
  Mathematically: For a center point $(x, y)$, the bounding box position can be represented as:

$$(x + \Delta x, y + \Delta y)$$

- **Objectness score:** To estimate if a bounding box indeed contains an object, an objectness score is predicted. This score is high for boxes containing objects and low for empty ones.
  Mathematically: The objectness score for a bounding box can be represented as $\sigma$, and $\sigma(x, y)$ represents the probability that the bounding box at $(x, y)$ contains an object.

- **Final bounding box prediction:** The final bounding box for an object is determined by adding the predicted offset to the center point. The size of the bounding box can also be a part of the prediction.
  Mathematically:
  The coordinates denoting the top-left and bottom-right corners of the bounding box can be expressed as follows:

  – Top-left corner:
  $$\left(x + \Delta x - \frac{w}{2}, y + \Delta y - \frac{h}{2}\right)$$

  – Bottom-right corner:
  $$\left(x + \Delta x + \frac{w}{2}, y + \Delta y + \frac{h}{2}\right)$$

Here, $(x, y)$ is the predicted center, $\Delta x$ and $\Delta y$ are the predicted offsets, and $(w, h)$ represents the width and height of the bounding box.

## IMPLEMENTATION AND EVALUATION

This section addresses the configuration of the model, the employed performance metrics, and the resulting assessment of performance.

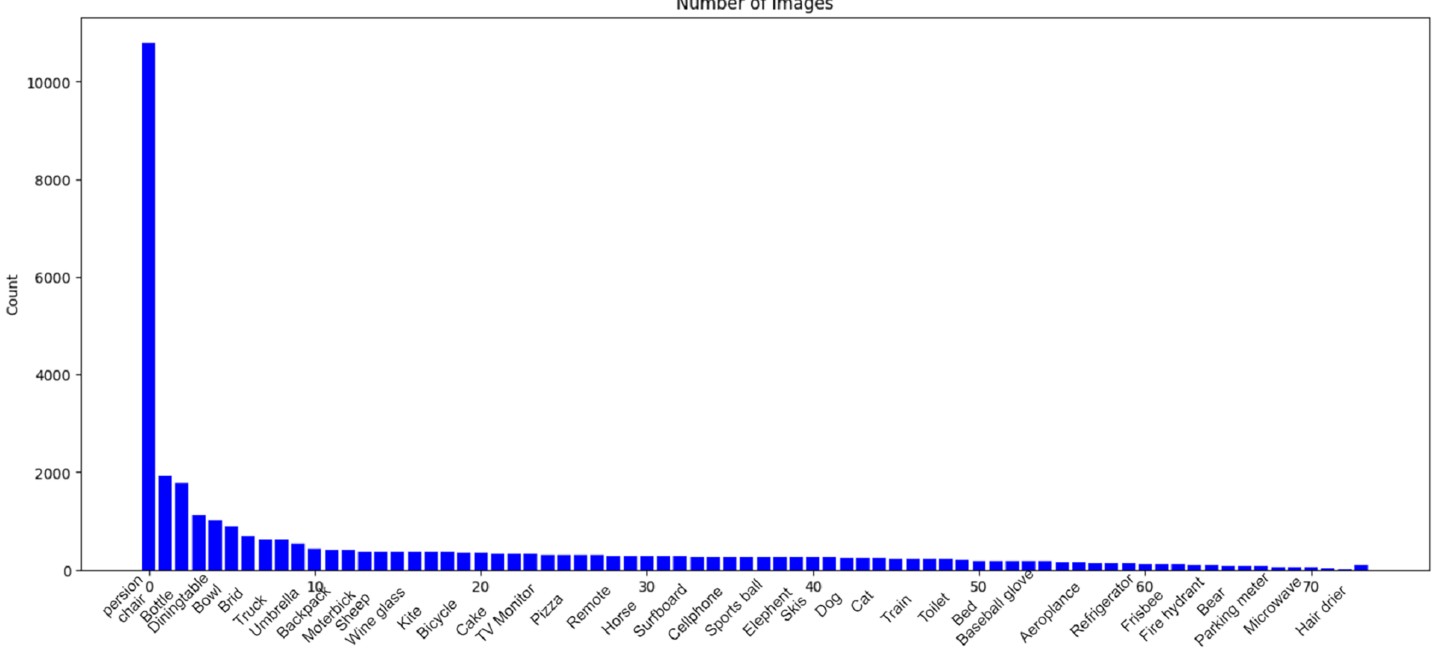

**Figure 5  Visual representation of object classes in the MSCOCO dataset.**

## Configuration

To facilitate comprehensive analysis in our research, we leverage the MS COCO dataset (*Lin et al., 2014*). The Common Objects in Context (COCO) dataset, developed by Microsoft and extensively documented, encompasses training, validation, and test sets comprising over 200,000 images distributed across 80 distinct object categories. This dataset serves as a comprehensive resource for object detection and image segmentation tasks. Figure 5 provides a visual representation of the various object classes present in the COCO dataset, offering a rich and diverse collection of images for training and evaluating computer vision models. Developed to advance object recognition and scene understanding, the COCO dataset has become a benchmark in the field, providing a standardized foundation for researchers and practitioners working on image-related tasks.

In configuring the model for our experiment, as illustrated in Table 1, we meticulously defined a set of parameters crucial to the training and performance of the neural network. The training process spanned 200 epochs, each representing a complete iteration through our dataset. To guide the optimization process, we employed a learning rate of 0.01, striking a balance between convergence speed and precision. Input images were standardized to dimensions of $512 \times 512$ pixels, and training occurred in batches of 16 images at a time, facilitating efficient parameter updates. Our comprehensive dataset, comprised of 26,520 images, served as the foundation for model learning.

The architecture, consisting of 225 layers, was meticulously chosen to capture intricate patterns in our data. With a total of 11,136,374 parameters, encompassing weights and biases, our configuration reflects a thoughtful balance between model complexity and

**Table 1 Configuration parameters.**

| Parameter | Value |
| --- | --- |
| Epoch | 200 |
| Image-size | 512 |
| Learning-rate | 0.01 |
| Batch-size | 16 |
| Layers | 225 |
| Number-of-images | 26,520 |
| Parameters | 11,136,374 |

computational efficiency, ultimately tailored to the specific demands of our research context.

## Performance metrics

The F-measure (*FM*) or F-1 score is determined by the weighted average of recall percentages and precision percentages. This score, also known as the harmonic mean of precision and recall, takes into account both false positives and false negatives. While *FM* is more commonly used than precision alone, accuracy is not immediately straightforward to comprehend. Accuracy performs well when false positives and false negatives have comparable costs. However, if the costs associated with false positives and false negatives differ, consideration of both recall and accuracy is preferable.

In the context of positive findings, precision (*P*) represents the proportion of accurately predicted observations to all predicted positive findings. On the other hand, recall (*R*) is the proportion of true positive predictions over all actual positives. These metrics can be calculated as follows:

$$P = \frac{TP}{TP + FP} \tag{3}$$

$$R = \frac{TP}{TP + FN} \tag{4}$$

In standard terminology, *TP* represents true positive, *FP* stands for false positive, *TN* denotes true negative, and *FN* refers to false negative. Precision and recall are taken into account when calculating the F-measure (*FM*), abbreviated as *FM* and computed using the following formula:

$$FM = \frac{2 \cdot R \cdot P}{R + P} \tag{5}$$

In addition to various other evaluation metrics, we incorporate the accuracy metric to evaluate the overall correctness of the model's predictions. The accuracy metric offers insight into the proportion of correctly classified predictions among the total number of object predictions. Its calculation is determined by the following formula:

$$\text{Accuracy} = \frac{TP + TN}{TP + TN + FP + FN} \tag{6}$$

Accuracy is a frequently employed performance metric, particularly in situations where the dataset is well-balanced, indicating a relatively equal number of positive and negative predictions. This metric provides a simple and direct measure to assess the correctness of the model in classification tasks. In addition to all other metrics, frame per seconds (FPS) is also used as the evaluation metric and model performance is judged accordingly.

Average precision (*AP*) is a widely utilized metric in object detection, evaluating a model's accuracy in detecting objects across different levels of precision. *AP* quantifies performance by computing the area under the precision-recall curve (*AUC − PR*) at various thresholds. The formula for calculating *AP* is expressed as:

$$AP = \sum_{i=1}^{n} (R_i - R_{i-1}) \cdot P_i \tag{7}$$

Here, *AP* represents the average precision, *n* denotes the number of data points, $R_i$ stands for recall at point *i*, $R_{i-1}$ represents recall at the previous point, and $P_i$ represents precision at point *i*.

Mean average precision (*mAP*) serves as an indicator of a model's comprehensive performance across all classes in object detection. It is determined by averaging the individual average precisions (*APs*) computed at various precision levels. The calculation of *mAP* can be expressed using the formula:

$$mAP = \frac{1}{N} \sum_{i=1}^{N} AP_i \tag{8}$$

In this formula, *mAP* represents the mean average precision, *N* denotes the number of classes, and *APi* signifies the average precision for each class *i*.

## Results and Discussion

Our IR-DD model training and testing were accelerated by the formidable NVIDIA RTX 2080Ti GPU and the Intel $I9 − 10900X$ CPU running at 3.7 *GHz*, resulting in substantial performance improvements.

Table 2 offers a comparative analysis of diverse object detection methods evaluated on the MS-COCO dataset (*Youssouf, 2022*; *Ye et al., 2023*; *Mahendru & Dubey, 2021*; *Guo et al., 2022*; *Qu et al., 2022*; *Dong, Tang & Zhang, 2023*; *Sirisha & Sudha, 2022*), presenting key performance metrics. The evaluation metrics, including precision, recall, F1 score, mean average precision (mAP) at IoU thresholds of 0.5 and 0.5:0.95, along with frames per second (FPS) during inference, offer a comprehensive assessment of each method's effectiveness. Notably, our proposed method achieves compelling results with a precision of 0.666, F1 score of 0.630, mAP@0.5 of 0.650, and an impressive FPS of 86. These metrics collectively underscore the competitive performance of our method in

**Table 2 Performance comparison of various methods.**

| Methods | Precision | Recall | F1 | mAP@0.5 | mAP@0.5:0.95 | FPS |
|---|---|---|---|---|---|---|
| Faster-RCNN | 0.529 | 0.607 | 0.565 | 0.513 | 0.220 | 29 |
| Cascade-RCNN | 0.494 | **0.656** | 0.564 | 0.548 | 0.250 | 24 |
| YOLOv3 | 0.608 | 0.612 | 0.610 | 0.627 | 0.308 | 48 |
| YOLOv4-CSP | 0.606 | 0.595 | 0.600 | 0.631 | 0.317 | 49 |
| YOLOv5 | 0.620 | 0.606 | 0.613 | 0.633 | 0.321 | 59 |
| YOLOv7 | 0.629 | 0.601 | 0.615 | 0.640 | **0.338** | 85 |
| CenterNet | 0.500 | 0.626 | 0.556 | 0.510 | 0.215 | 70 |
| **Ours** | **0.666** | 0.608 | **0.630** | **0.650** | 0.335 | **86** |

Note:
Best values are bold in each criterion.

**Table 3 Performance results for different versions of YOLOs.**

| YOLO Version | References | FPS | AP50 |
|---|---|---|---|
| YOLOv3 | *Redmon & Farhadi (2018)* | 20 | 57.9 |
| YOLOv4 | *Bochkovskiy, Wang & Liao (2020)* | 62 | 65.7 |
| YOLOv5-L | *Shafiee et al. (2017)* | 113 | 67.3 |
| PP-YOLO | *Long et al. (2020)* | 73 | 65.2 |
| PP-YOLOV2 | *Huang et al. (2021)* | 50.3 | 69 |
| YOLOv6-L | *Li et al. (2022)* | 98 | 70 |
| YOLOv7 | *Wang, Bochkovskiy & Liao (2023)* | 161 | 69.7 |
| **YOLOv8** | – | **280** | **53.9** |

Note:
Best values are bold in each criterion.

object detection, demonstrating proficiency across precision, recall, F1 score, mAP, and computational efficiency on the MS-COCO dataset.

Table 3 and Fig. 6 provides a comparative performance overview for various iterations of the YOLO object detection algorithm, evaluating their effectiveness through two key metrics: frames per second (FPS) and average precision at an IoU threshold of 0.5 (AP50). YOLOv8 emerges as the fastest, boasting an impressive 280 FPS; however, this efficiency comes at a cost, as it registers the lowest AP50 at 53.9. In contrast, YOLOv6-L strikes a notable balance between speed and accuracy, achieving the highest AP50 of 70 while maintaining a respectable FPS of 98. YOLOv5-L also performs competitively, with an FPS of 113 and an AP50 of 67.3. These metrics offer practitioners valuable insights, emphasizing the nuanced trade-off between speed and precision when selecting the most suitable YOLO version for specific object detection applications. Notably, the utilization of the YOLOv8 model in our road distress detection system is strategically grounded in its superior performance metrics, striking a well-calibrated balance between rapid processing capabilities and precise object localization.

The Fig. 7 describes a collection of illustrative instances highlighting the results of object detection using our model on the MS-COCO dataset (*Lin et al., 2014*). These instances, presented in the form of figures, showcase the model's performance in accurately detecting

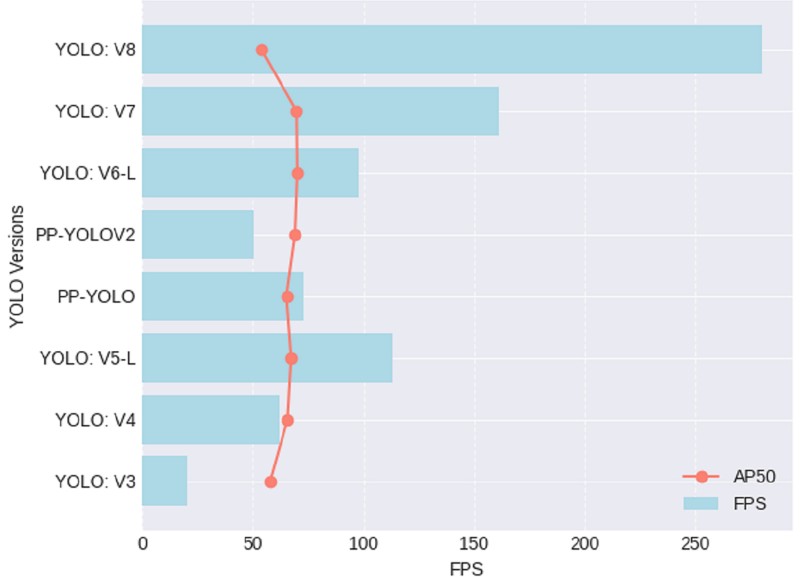

**Figure 6  Performance results for different versions of YOLOs.**

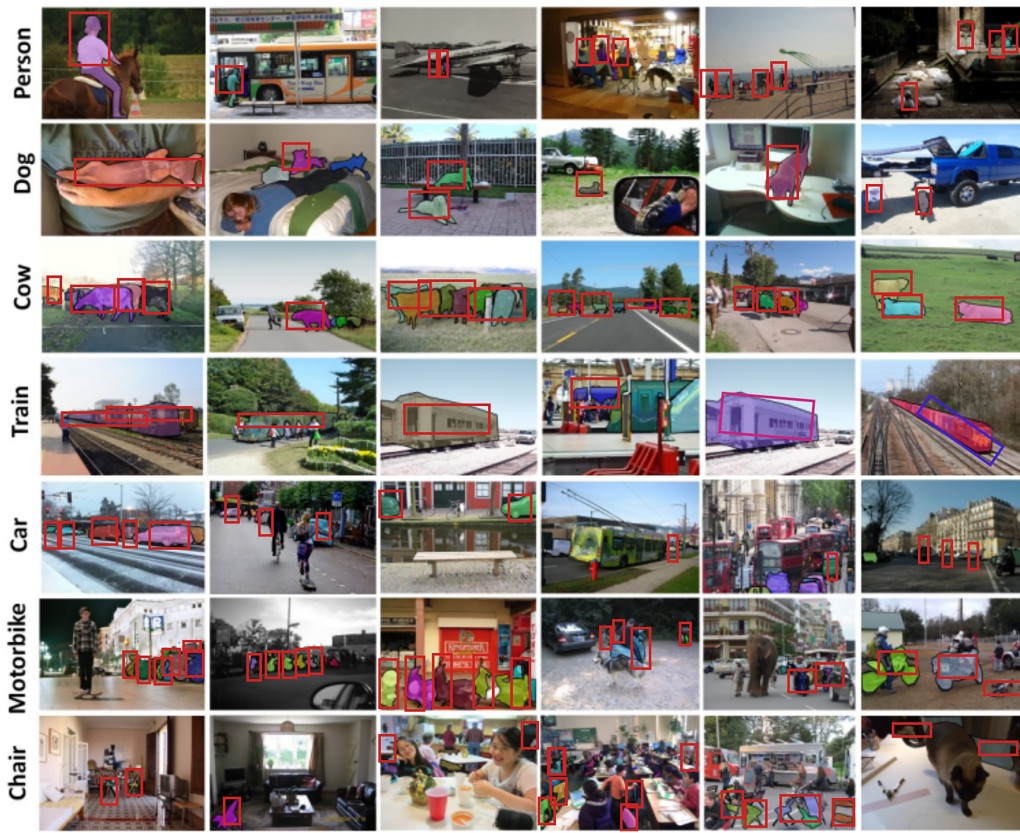

**Figure 7  Several illustrative instances showcasing the outcomes of our object detection on the MS-COCO dataset.** Image credit: *Lin et al. (2014)* Copyright © 2014, Springer International Publishing Switzerland.

and delineating various objects within different pictures. The use of YOLOv8, known for its efficiency in real-time object detection, emphasizes the effectiveness of the model in identifying and localizing objects across diverse scenarios present in the MS-COCO dataset. These visual representations serve as concrete examples of the model's capabilities and contribute to a more tangible understanding of its object detection performance.

In our study, the integration of the EfficientNet as a backbone, coupled with the utilization of BiFPN, has proven to be a strategically advantageous choice for enhancing the performance of our model. EfficientNet, known for its superior scaling properties and efficient use of model parameters, serves as a robust foundation for feature extraction, enabling the network to capture intricate patterns and representations within the input data. The incorporation of BiFPN further contributes to improved feature integration and information flow across different network layers, fostering enhanced contextual understanding. The bidirectional connections in BiFPN facilitate effective communication between feature maps at various scales, promoting richer feature representations crucial for accurate object detection. The synergistic combination of EfficientNet and BiFPN, as demonstrated in our experimental results, not only optimizes model efficiency but also bolsters the overall precision and recall of our object detection system, making it well-suited for diverse and complex visual scenarios. This strategic architectural choice underscores the significance of leveraging state-of-the-art components to achieve superior performance in computer vision tasks.

## CONCLUSION AND FUTURE WORKS

In conclusion, our proposed framework, leveraging the YOLOv8 algorithm and incorporating BiFPN for recursive feature fusion and bidirectional connections, has demonstrated notable advancements in road distress detection. The synergy between YOLOv8's anchor-free architecture and BiFPN's multi-scale feature optimization has significantly improved accuracy and real-time capabilities. Through comprehensive experimental analysis, we have shown that our approach surpasses traditional methods, making it a cost-effective alternative for practical applications such as smart cities and autonomous vehicles. The iterative refinement of features and bidirectional information exchange contribute to a more robust and efficient system for identifying road anomalies.

In future research, it is crucial to explore newer versions or variations of the YOLO series to assess their effectiveness in road distress scenarios. Balancing speed and accuracy, especially in diverse distress complexities, requires further investigation to address limitations such as low AP50. Overcoming challenges related to detecting smaller objects and refining the framework for diverse datasets is essential. The adaptability of the proposed system to real-world environmental factors and different geographic regions presents promising avenues for exploration. Additionally, integrating advanced anomaly detection algorithms and considering temporal aspects in video-based distress detection could significantly enhance the system's overall efficacy. This study establishes a foundation for ongoing advancements in road distress detection, serving as a launchpad for more sophisticated and robust applications in the future.

## Funding

This work was supported by the School-level project of SZPU Grants (No. 6021330010K and No. 6023310026K), the Post-doctoral Later-stage Foundation Project of Shenzhen Polytechnic (No. 6021271014K) and the Shenzhen Polytechnic Scientific Research Start-up Project (6022312028K). The funders had no role in study design, data collection and analysis, decision to publish, or preparation of the manuscript.

## Grant Disclosures

The following grant information was disclosed by the authors:
School-level project of SZPU: 6021330010K and 6023310026K.
Post-doctoral Later-stage Foundation Project of Shenzhen Polytechnic: 6021271014K.
Shenzhen Polytechnic Scientific Research Start-up Project: 6022312028K.

## Competing Interests

The authors declare that they have no competing interests.

## Author Contributions

- Ahsan Zaman Awan conceived and designed the experiments, performed the experiments, analyzed the data, performed the computation work, prepared figures and/or tables, authored or reviewed drafts of the article, and approved the final draft.
- Jiancheng (Charles) Ji analyzed the data, authored or reviewed drafts of the article, and approved the final draft.
- Muhammad Uzair conceived and designed the experiments, performed the experiments, prepared figures and/or tables, and approved the final draft.
- Irshad Ullah performed the experiments, authored or reviewed drafts of the article, and approved the final draft.
- Waqar Riaz performed the computation work, prepared figures and/or tables, and approved the final draft.
- Tao Gong performed the computation work, authored or reviewed drafts of the article, and approved the final draft.

## Data Availability

    The MS COCO dataset is available at https://datasetninja.com/coco-2017.
    The code and raw data are available in the Supplemental Files.

## Supplemental Information

Supplemental information for this article can be found online at http://dx.doi.org/10.7717/peerj-cs.2038#supplemental-information.

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
