# Peer review of "Innovative road distress detection (IR-DD): an efficient and scalable deep learning approach"

_PeerJ Computer Science, doi:10.7717/peerj-cs.2038_

## Round 0.1 · original submission · Major Revisions

Clarify the main contributions of this research work and include some relevant papers that were published recently.

**Language Note:** PeerJ staff have identified that the English language needs to be improved. When you prepare your next revision, please either (i) have a colleague who is proficient in English and familiar with the subject matter review your manuscript, or (ii) contact a professional editing service to review your manuscript. PeerJ can provide language editing services - you can contact us at [email protected] for pricing (be sure to provide your manuscript number and title). – PeerJ Staff

·

Basic reporting

After reading this article, I have the following suggestions and questions:. The paper needs to improve according to theses.
1) The abstract could be enhanced by explicitly stating the main problem addressed and the unique aspects of your proposed solution.
2) The manuscript contains several English-language errors. It is advised to conduct thorough proofreading to rectify these issues.
3) Briefly specify the challenges addressed by IR-DD for more clarity.
4) Certain sections of the manuscript could benefit from clearer expressions. Please review and enhance the clarity of your statements.
5) The resolution of Figure 4 appears to be suboptimal. Kindly consider redrawing it for improved clarity and visual quality.
6) Explain clearly the integration of YOLOv8 and BiFPN in the IR-DD framework.
7) Ensure that all in-text citations are accurately linked to the corresponding references in the bibliography.
8) Revised the abstract and briefly specified the challenges addressed by IR-DD for more clarity.
9) The use of YOLOv8 and BiFPN in IR-DD, it could benefit from emphasizing the novelty of this integration. Clearly articulating how this combination is unique and contributes to the field of road distress detection.
10) Consider adding a brief section on potential future work or extensions of IR-DD, providing readers with insights into the ongoing research possibilities.
11. reviewed and compared your work with research "Islam, U., Al-Atawi, A. A., Alwageed, H. S., Mehmood, G., Khan, F., & Innab, N. (2024). Detection of renal cell hydronephrosis in ultrasound kidney images: a study on the efficacy of deep convolutional neural networks. PeerJ Computer Science, 10, e1797."

Experimental design

Nil

Validity of the findings

Nil

Additional comments

Nil

Reviewer 3 ·

Basic reporting

The article must be written in English and must use clear, unambiguous, technically correct text. The article must conform to professional standards of courtesy and expressio

Experimental design

Original primary research within Aims and Scope of the journal.

Validity of the findings

GOOD

Additional comments

no more

Reviewer 4 ·

Basic reporting

1.Enhance the impact of Section 1 by explicitly delineating the primary problem addressed in paragraph-3.
2.Add more recent studies to make sure you're covering the latest research in the field.

Experimental design

no comment

Validity of the findings

Would you mind providing more details about what makes your approach or contribution unique?

Additional comments

Enhance the manuscript's overall coherence for a more seamless flow of ideas

·

Basic reporting

1.Enhance clarity by briefly elucidating the challenges confronted by the proposed approach.

Experimental design

2.Refine the expressions in section 3.4 line 307 to 313 and section 5 line 443 to 450 of the manuscript for improved clarity.

Validity of the findings

3.Clarify the BiFPN integration by explaining (section 3.3) how this combination in distinctive and contributes uniquely to the field of road distress detection?

---

## Round 0.2 · accepted · Accept

The paper is revised according to the reviewers' comments.

·

Basic reporting

The author addressed my concerns, so i accepted it for publication.

Experimental design

Nil

Validity of the findings

Nil

Reviewer 3 ·

Basic reporting

It's good paper over all paper written good manner.

Experimental design

Designing novelty is very good

Validity of the findings

No comment

Additional comments

It's not required